# Anti-SARS-CoV-2 Strategies and the Potential Role of miRNA in the Assessment of COVID-19 Morbidity, Recurrence, and Therapy

**DOI:** 10.3390/ijms22168663

**Published:** 2021-08-12

**Authors:** Maria Narożna, Błażej Rubiś

**Affiliations:** 1Department of Pharmaceutical Biochemistry, Poznan University of Medical Sciences, 4 Święcickiego St., 60-781 Poznan, Poland; maria.narozna@ump.edu.pl; 2Department of Clinical Chemistry and Molecular Diagnostics, Poznan University of Medical Sciences, 49 Przybyszewskiego St., 60-355 Poznan, Poland

**Keywords:** coronavirus, SARS-CoV-2, COVID-19, miRNA, miR, COVID-19 therapy

## Abstract

Recently, we have experienced a serious pandemic. Despite significant technological advances in molecular technologies, it is very challenging to slow down the infection spread. It appeared that due to globalization, SARS-CoV-2 spread easily and adapted to new environments or geographical or weather zones. Additionally, new variants are emerging that show different infection potential and clinical outcomes. On the other hand, we have some experience with other pandemics and some solutions in virus elimination that could be adapted. This is of high importance since, as the latest reports demonstrate, vaccine technology might not follow the new, mutated virus outbreaks. Thus, identification of novel strategies and markers or diagnostic methods is highly necessary. For this reason, we present some of the latest views on SARS-CoV-2/COVID-19 therapeutic strategies and raise a solution based on miRNA. We believe that in the face of the rapidly increasing global situation and based on analogical studies of other viruses, the possibility of using the biological potential of miRNA technology is very promising. It could be used as a promising diagnostic and prognostic factor, as well as a therapeutic target and tool.

## 1. The Challenge

In Wuhan city in China, pneumonia cases were reported at the end of 2019 [1]. The ongoing disease was referred to as COVID-19 (coronavirus disease 2019), causing the severe acute respiratory syndrome coronavirus 2 (SARS-CoV-2). COVID-19 became a pandemic and spread all over the world. The World Health Organization (WHO) reported 202,538,813 confirmed cases and 4,292,637 COVID-19-related deaths globally as of 7 August 2021 [2]. It is not easy to find a link between susceptibility to COVID-19 and the molecular characteristics of individual patients. Thus, finding a reliable morbidity and recurrence marker or therapy target is a challenging task. The standard procedure to prevent a viral infection, i.e., vaccination, is a promising strategy. However, both the issue of appearing mutants [3] and the efficacy and durability of immunity after vaccination remain a riddle. This relatively novel virus also has a too-short history of coexistence with humans to provide a thorough analysis. However, since it belongs to the coronaviruses’ family, it might be possible to find some analogy to other viral infections and identify a potential therapy direction. 

Now, while immunization programs in the world are in progress, studies are running in the area of in silico [4], in vitro [5], and in vivo [6] research. Naturally, most of the efforts are focused on identifying target points that would provide efficient prevention or therapy. Consequently, the number of ongoing clinical trials is increasing, detailing current drug application principles and their potential negative adverse effects. Until now (August 2021), the only treatment therapy that the FDA has approved is the antiviral drug remdesivir, approved for use in adults and pediatric patients for treatment requiring hospitalization [7]. Besides that, the FDA authorized the REGEN-COV (casirivimab and imdevimab, administered together) and sotrovimab for the treatment of certain patients with COVID-19 [8]. 

It is even more challenging as we know that SARS-CoV-2 exhibits a highly mutable single-stranded RNA genome [9]. Currently, few vaccine systems have been reported (for details, see below and Figure 1), making the vision of pandemic elimination more real. However, some other similar threats may appear in the future. Therefore, we require an evaluation of a systemic approach in similar situations, especially as globalization proceeds and a significant increase in human population is expected.

Noteworthily, the SARS-CoV-2 messenger RNA-based vaccines were the first authorized for use accessible to the worldwide public in December 2020 [10]. The reports concerning vaccines created by, among others, Moderna, BioNTech/Pfizer, Oxford/AstraZeneca, and Johnson & Johnson have shown promising results in activating the immune system against the virus. Therefore, they are already being applied globally [11]. 

Spreading variants of the SARS-CoV-2 coronavirus are growing worldwide [12]. The variants identified in the UK (B.1.1.7 and B.1.1.7+E484K [13]), South Africa (B.1.351), South America (P.1), and India (B.1.617) [14] are highly transmissible [15]. These variants, especially the Indian strain, not only replaced the previously dominant strains in their regions, but most of all, they severely endanger health worldwide. Epidemiologists studying the outgrowth of SARS-CoV-2 variants in the UK estimated the spreading rate of the current British variant to possess transmissivity 70% greater than others [16]. Consequently, novel diagnostic and therapeutic strategies should be proposed. 

Thus, in the face of the rapidly increasing global situation, and based on the analogical studies of other viruses, we raise the possibility of using the biological potential of miRNAs as promising diagnostic and prognostic factors, as well as therapeutic targets and tools.

## 2. Knowing the Enemy

The angiotensin-converting enzyme 2 (ACE-2)-rich epithelial cells in the nasal cavity and alveoli are the main entry sites for SARS-CoV-2 [17]. Thus, COVID-19 is viewed as a disease that mainly affects the lungs, but it can easily damage other organs. Based on single-cell RNA sequencing datasets, Zou et al. [18] mapped the risk of organ susceptibility to the virus invasion based on ACE-2 expression, indicating the airways, lungs, heart, kidneys, intestines, and other organs [18]. Hence, pulmonary disorders were the most frequently noted, as the literature describing the extrapulmonary effects of COVID-19 is ample [19]. The entry of SARS-CoV-2 into cells via membrane fusion leads to the downregulation of ACE-2 and the consequent loss of their catalytic ability to degrade angiotensin II. Some symptoms, such as pneumonia and coagulation, are partly due to elevated angiotensin II [20]. They may also result from the widespread replication of the virus due to endothelial cell damage, thrombosis-related inflammation, and deregulation of the immune response (i.e., the so-called "cytokine storm syndrome" (CSS)) [21]. 

Pneumonia often associated with COVID-19 can cause long-term damage to the alveoli. The damage caused by the disease can lead to long-term breathing obstacles. Even in young people, COVID-19 can generate strokes, seizures, and Guillain–Barre syndrome, a condition that causes temporary paralysis. Examinations taken months after recovering from COVID-19 showed permanent damage to the heart muscle, even in people who only experienced mild symptoms [22]. It might increase the risk of heart failure or other cardiac complications in the future. In addition, COVID-19 may further increase the risk of developing Parkinson’s disease, Alzheimer’s disease, and the formation of blood clots [23,24]. Since large clots can cause heart attacks and strokes, most heart injuries from COVID-19 are believed to come from tiny clots that block the heart muscle capillaries. The disease can also weaken blood vessels and cause them to leak, contributing to potentially long-term problems with liver and kidneys’ metabolism [23]. Many people who have recovered from SARS-CoV-2 have developed chronic fatigue syndrome, worsening physical or mental activity, and not elapsing with rest. Although, many of the long-term effects of COVID-19 remain unknown [25].

Notably, the outcomes of asymptomatic infections are a relevant area of prospective research. Such patients are much less likely to be monitored for long-term sequelae, while current knowledge already presents evidence that such sequelae exist [26]. Furthermore, the severity of the current pandemic and the numerous health consequences of infection indicates the necessity to identify innovative therapeutic strategies [27]. Consequently, various approaches focus on COVID-19 treatment and prevention. Potential drug candidates are being implemented based on virulence mechanisms and pathophysiology [28] (Figure 2). These first start from the impact on the life cycle of SARS-CoV-2 and its stages, such as host cells’ binding (mainly via ACE-2 and TMPRSS2 receptors and the Spike protein) [29], endocytosis, viral replication (at the stage of translation and replication RNA virus) [30]. Further, they analyze and influence the virus’s pathogenicity by controlling the inflammatory responses that cause moderate to severe COVID-19 disease [31]. Finally, they study possible effective therapeutic mechanisms of different classes of drugs that have been or are currently being tested in preclinical and clinical trials (also in other viral infections).

Some key proposed treatment strategies are drugs inhibiting viral entry (A) or replication (B) and activating internal immunity and inflammatory responses (C) [32]. Drugs interacting with RAAS are ACEI, ARBs, and specific human antibodies. The SARS-CoV-2 attachment mechanism (panel A) is as follows: The S protein binds ACE-2, allowing entry into the host cell and infection; ACE converts angiotensin I to angiotensin II; Angiotensin II exerts biological functions through AT1R and AT2R, leading to severe vasoconstriction in organs; ACE-2 hydrolyzes angiotensin II to the vasodilator angiotensin 1-7, which binds to the Mas receptor and has a protective function in several organs. TNF-α, along with cytokines released from SARS-CoV-2 infection, may lead to CSS [33]. While the S protein binds ACE-2 receptors, the activity of proteases such as TMPRSS2 plays a crucial role. SARS-CoVs may need TMPRSS2 to enter the host cell as it plays a pivotal role in activating the S protein for membrane fusion. Therefore, TMPRSS2 inhibitors may also be part of COVID-19 therapy [34]. Other drug candidates can target components activating CSS, such as the JAK-STAT inhibitors, pro-inflammatory cytokine antagonists (against IL-1, IL-6), TNF-α blockers, and steroids (panel C). SARS-CoV-2 inhibits the IFN type I receptor’s activation, which reduces this antiviral response mechanism while allowing the pro-inflammatory IL-1, IL-6, activating CSS [35]. The clathrin-dependent endocytosis of SARS-CoV-2 requires proteins such as PICALM, which is necessary for endosome maturation. Chloroquine and hydroxychloroquine may reduce PICALM [36] and inhibit the clathrin-mediated endocytosis (panel A), which was proven during treatment combined with lopinavir/ritonavir [36]. These two agents disrupt viral replication. Potential methods of interference with viral replication can be direct RdRp inhibition that interrupts RNA replication (such as HCV inhibitors) and the application of nucleoside analogs causing premature termination of RNA replication and indirectly inhibiting the activity of RdRp [37] (panel B). Oxidative stress inhibitors may also be useful in treating COVID-19. Vitamin C, an antioxidant, and selenium, which supports a group of enzymes, may contribute to the prevention of ROS and cell damage [38]. Research shows that COVID-19 patients may also have a vitamin D (membrane antioxidant) deficiency [39], which was further proven to inhibit ACE-2 expression [40], thereby it may limit SARS-CoV-2 cell entry (panel C). 

## 3. Potential Therapeutic Strategies towards COVID-19

Since the new coronavirus is highly infectious and lethal, it is of great importance to recognize the metabolic and replicative pathways of SARS-CoV-2 [41]. Knowing those facts would increase the chances of identifying potential prognostic markers and targets that would provide more efficient pandemic extinguishing. So far, we are focused on two approaches, i.e., looking for analogy in other pandemics and using the drugs/strategies that already exist or identifying novel ones [42]. However, apart from a pretty obvious solution, i.e., vaccination (will be addressed further), there are still some alternative approaches that we consider. 

SARS-CoV-2 genome is translated into 29 structural and nonstructural proteins, including spike (S), membrane (M), envelope (E), and nucleocapsid (N) proteins, the RNA-dependent RNA polymerase (RdRp), and the papain-like protease (PLpro) [43], which have been recognized as potential immunizing agents as well as targets for inhibitors that are currently available and used in various viral therapies. The inhibition of the protease activity was proposed against COVID-19 infection [44]. Interestingly, protease inhibitors (PIs) like lopinavir, the HIV-1 protease inhibitor, and ritonavir, a CYP3A4 inhibitor, demonstrated in vitro antiviral efficacy against SARS and MERS through the inhibition of the 3-chymotrypsin-like protease [45]. However, in the randomized, controlled, open-label platform trial conducted by Horby et al. from the RECOVERY Collaborative Group, lopinavir–ritonavir treatment was not associated with reducing 28-day mortality in hospitalized patients COVID-19 patients or the prospect of progressing to invasive mechanical ventilation or death [46]. 

A broad-spectrum prodrug, adenosine analog, and an RdRp inhibitor, remdesivir (RDV), showed efficacy in targeting SARS-CoV-2 replication in animal models [47]. Owing to its nucleoside analog activity, RDV can drive premature viral RNA chain termination. However, Grein et al. [48] reported side effects after using RDV in patients with COVID-19. The symptoms included diarrhea, a rash, hypotension, liver function abnormalities, and renal dysfunction. Serious side effects, such as acute kidney damage, septic shock, and multi-organ failure, were diagnosed in 23% of patients. Another clinical study proved that convalescence plasma therapy with RDV became beneficial [49]. Eventually, in a randomized trial, RDV was not associated with statistically significant clinical benefits. Further studies should provide additional data on its efficacy [50].

In August 2021, REGEN-COV (formerly known as REGN-COV2), a combination of two monoclonal antibodies (casirivimab and imdevimab), was confirmed to significantly reduce the risk of hospitalization or death among people at high risk of contracting COVID-19. Subcutaneous REGEN-COV prevented symptomatic and asymptomatic infection in previously uninfected individuals who had been in contact with infected individuals. Among participants who became infected, REGEN-COV reduced the duration of symptomatic disease and the duration of the high viral load [51].

### 3.1. Old Antimalarial Drugs

Two specific drugs, chloroquine and hydroxychloroquine, exhibiting anti-inflammatory, antiproliferative, and immunomodulating capacities, were also included in clinical trials. They were proven to modulate the glycosylation of the ACE-2 receptor [52]. Hydroxychloroquine, an old antimalarial drug, inhibited coronavirus replication in vitro [53]. Since the virus has been proven to utilize the ACE-2 receptors, it has been hypothesized that hydroxychloroquine may also interfere with its glycosylation, thus preventing SARS-CoV-2 binding to its target cells. Additionally, hydroxychloroquine could inhibit lysosomes’ acidification and interfere with the fusion process of the SARS-CoV-2 with the host cell [54]. However, evidence provided insufficient data supporting its efficacy for all safety outcomes [54]. Another drug candidate, azithromycin, is a broad-spectrum macrolide derivative of erythromycin. It has also shown in vitro antiviral activity against SARS-CoV-2 after treatment with similar doses to those used in bacterial pneumonia treatment [55]. It probably interferes with the acidification processes of lysosomes and endosomes or enhances interferon antiviral activity. Azithromycin combined with chloroquine/hydroxychloroquine has been proposed as a COVID-19 treatment option [55]. Although clinical trials by Abaleke et al. [56] confirm that azithromycin treatment in COVID-19 patients admitted to the hospital did not improve survival or other prespecified clinical outcomes, and its usage should be limited only to an antimicrobial indication. An FDA summary on hydroxychloroquine and chloroquine in the treatment of COVID-19 patients is also available. It includes reports of severe side effects, including heart problems and other safety concerns, comprising blood and lymphatic disorders, kidney injuries, and liver failures [57]. Moreover, in an open-label platform study, the Horby team randomized 1561 patients to receive hydroxychloroquine and 3155 to ordinary care. The primary endpoint was mortality at 28 days. Patients who received hydroxychloroquine did not show a lower death rate after 28 days than those who received usual care [58].

### 3.2. Investigation into the Role of Vitamins

The role of vitamin C in preventing and treating pneumonia and sepsis has been raised for many decades [59]. These studies provided the basis for studies of patients with severe symptoms of COVID-19. Patients with pneumonia and sepsis have low levels of vitamin C and elevated levels of oxidative stress. Hence, administering vitamin C to patients with pneumonia may reduce the symptoms’ severity and duration of the disease [60]. Critically ill patients with sepsis require high doses of intravenous vitamin C to normalize plasma levels. Vitamin C has pleiotropic physiological functions, many of which appear to be related to COVID-19 susceptibility, including antioxidant, anti-inflammatory, anticoagulant, and immunomodulatory properties [61]. Preliminary studies show low vitamin C levels in critically ill COVID-19 patients [62]. Among the integrative therapies, vitamin C infusion may increase the synthesis of norepinephrine and vasopressin, reduce cytokine levels, and prevent neutrophils’ activation [63]. In the study by Hiedra et al. [64], 17 patients with COVID-19 received 1 g of vitamin C intravenously for three days. These patients were initially treated with hydroxychloroquine, methylprednisolone, or tocilizumab (a monoclonal antibody competitively inhibiting the binding of interleukin-6 to its receptor (IL-6R)). After vitamin C therapy, the levels of some anti-inflammatory markers, such as ferritin, were significantly reduced. However, this study lasted only three days and did not investigate the effects of vitamin C alone. Several clinical trials are now ongoing in COVID-19 patients using vitamin C as an intravenous or oral treatment. For example, a clinical trial in Wuhan, China, is investigating the role of vitamin C infusions for the treatment of severe COVID-19-related pneumonia cases [65]. 

The efficacy of interleukin-6 receptor antagonists such as tocilizumab or sarilumab alone in critically ill COVID-19 has also been tested. Gordon et al. introduced tocilizumab and sarilumab in an international multivariate adaptive study. COVID-19 patients were randomized to receive tocilizumab (8 mg per kilogram body weight), sarilumab (400 mg), or standard of care within 24 hours of starting organ support in the intensive care unit. In critical patients receiving organ support, treatment with interleukin-6 receptor antagonists improved outcomes, including survival [66]. Abani et al. also confirmed that tocilizumab improved survival in hospitalized patients with hypoxia and systemic inflammation. Patients assigned to the tocilizumab treatment were less prone to reach the composite endpoint of invasive mechanical ventilation or death [67].

The possible link between vitamin D3 (VD3) deficiency and susceptibility to SARS-CoV-2 has also recently gained attention from the medical community. There are molecular pathways in which VD3 regulates the processes involved in both SARS-CoV-2 replication and cell survival [68]. Several reports indicate that VD3 deficiency is associated with a higher risk of upper respiratory tract infection [69]. In the respiratory syncytial virus, VD3 increases the synthesis of the IκBα (NF-κB inhibitor) in the respiratory epithelium, resulting in a reduction in the expression of pro-inflammatory genes. It was demonstrated by silencing the vitamin D receptor (VDR) with siRNA accompanied by no decrease in the expression of the pro-inflammatory genes [70]. Moreover, single nucleotide polymorphisms in the VDR are associated with more severe sequelae of the respiratory syncytial virus [70]. A clinical trial that involved more than four thousand COVID-19 patients found that vitamin D deficiency was associated with an almost two-fold risk of contracting the virus. The effective dose of VD3 to provide at least partial protection against the effects of SARS-CoV-2 infection is unknown and is under investigation. Serum VD3 levels are a delayed marker of whole-body VD3 status as tissue reserves are depleted before serum VD3 levels fall below normal [71]. Clinical trials of oral vitamin D supplementation in COVID-19 patients are registered at clinicaltrials.gov, for example, randomized trial of high-dose versus standard-dose VD3 in high-risk COVID-19 patients [72]. Vitamin D deficiency may be a common variable amongst the elderly and patients with underlying diseases in populations at higher risk of complications and mortality from COVID-19 [73]. It might justify the use of vitamin D as a maintenance and expression agent in the immune response, which is essential in decreasing the risk and severity of the viral infection and alleviating the symptoms of the disease [74].

Further research will also emerge on an appropriate loading dose for VD3-deficient patients to improve COVID-19 treatment and prevention due to the antioxidant role similar to vitamin C and its contribution to other mechanisms. Nevertheless, there are no precise recommendations regarding the dosage of vitamin D supplementation for the prevention or treatment of COVID-19 [75]. 

### 3.3. Convalescent Plasma Transfusions

Another approach was to exploit the efficacy of convalescent plasma (CP) transfusion as a promising therapy in viral infections. CP has been used for over a century as post-exposure prophylaxis and treatment of various infectious diseases. Cases from prior viral outbreaks suggested that it reduced viral load and cytokine levels, improving clinical outcomes [76]. Even so, the randomized clinical trials of Simonovich et al. [77] reported that CP did not help fight the symptoms of COVID-19. A total of 228 patients received CP, and 105 received a placebo. There were no significant differences in the clinical outcomes between the group that received CP and the group that received a placebo. Total mortality was 10.96% in the CP group and 11.43% in the placebo group. Hence, no significant clinical differences or overall mortality were observed between plasma-treated patients and those who received a placebo [77]. Additionally, the safety and efficacy of CP therapy in patients were also assessed through a randomized, controlled, open-label platform study by Abani et al. [78]. In patients hospitalized for COVID-19, high-titer convalescent plasma did not improve survival or other predetermined clinical outcomes.

### 3.4. In Response to the Cytokine Storm Syndrome: Immunomodulation Effects in COVID-19 Management

Evidence suggests that COVID-19-related deaths are mainly due to the increased inflammation. Therefore, the intensity of pro-inflammatory cytokines releases significantly differentiates between mild and severe COVID-19 cases [79]. 

CSS and organ damages characterized by an excessive release of pro-inflammatory cytokines might be reduced by immunomodulatory drugs. Corticosteroids, such as methylprednisolone and dexamethasone, significantly reduced the death rate in patients with COVID-19 with acute respiratory distress syndrome (ARDS). A clinical study showed that methylprednisolone reduced the risk of death in patients with ARDS [80]. Other studies found that a single dose of 40–500 mg of methylprednisolone could diminish the inflammatory process without adversely affecting viral clearance and specific IgG production. However, long-term administration of methylprednisolone delayed viral clearance and suppressed the immune system [81]. Dexamethasone is also an approved, potent corticosteroid with predominantly glucocorticoid action exhibiting anti-inflammatory and immunosuppressive effects. In the studies conducted by the UK Recovery group led by Horby [82], patients were treated with oral or intravenous dexamethasone (6 mg/day) for up to 10 days or to receive the usual care alone. The primary endpoint was mortality at 28 days. More than 2000 patients were assigned to receive dexamethasone and more than 4000 to regular care. In the dexamethasone group, the death rate was lower than in patients receiving invasive mechanical ventilation and those receiving oxygen without invasive mechanical ventilation. Thus, they provided evidence that in hospitalized patients for COVID-19, the use of dexamethasone resulted in a lower mortality rate among those receiving either invasive mechanical ventilation or oxygen alone [82].

Janus kinase inhibitors also exhibit antiviral effects by blocking endocytosis and cytokines activation [83]. Baricitinib, for example, is a clathrin-mediated viral endocytosis regulator, preventing SARS-CoV-2 penetration into the alveolar epithelial cells. An open-labeled study performed by Cantini et al. [84] showed moderate COVID-19 pneumonia patients treated with baricitinib combined with lopinavir–ritonavir improved in, most of all, fever, oxygen saturation, and C-reactive protein. Furthermore, 1033 COVID-19 patients were also subjected to a randomized trial, of which the results revealed that baricitinib administered with RDV showed better clinical outcomes than RDV alone by accelerating improvement and reducing recovery time [85].

## 4. Vaccination Strategy

### 4.1. The Vaccine Run

To date, emerging options in vaccination strategies are under development, including inactivated viral vaccines, live-attenuated viral vaccines, viral vector vaccines, viral subunit vaccines, and viral nucleic acid vaccines [86] (Figure 1). Many clinical trials have started, and as of May 29, 2021, 16 vaccines were already authorized and approved for use in different regions of the world [87]. Still, identifying a new, specific, and efficient vaccine brings other challenges, i.e., the rapid large-scale manufacturing and overcoming the issue associated with novel mutated variants of the virus (Figure 1). The currently available vaccines are most likely to protect against the dominant SARS-CoV-2 variants, but we still need more scientific evidence. Recently, a new, dangerous mutant variant was identified in India and, as of May 10, 2021, already spread to more than 30 countries [88]. Animal studies indicate that it can cause a much more severe disease [89]. Yadav et al. showed that hamsters infected with B.1.617 had more lung inflammation than animals infected with other variants [90]. Unfortunately, the currently used vaccines may not be so effective in this case. Ferreira et al. [91] suggests that the antibodies are less effective against the Indian variant. The antibodies produced by people vaccinated with Pfizer were about 80% effective against some mutations in B.1.617. However, they emphasize that this does not make the vaccination ineffective [91]. Similarly, the team of Hoffmann et al. [92] tested the sera of people previously infected with SARS-CoV-2 and found that their antibodies neutralized B.1.617 about 50% less effectively than previously circulating strains. In addition, when they tested the serum of patients after two doses of the Pfizer vaccine, they found that the antibodies were about 67% effective against variant B.1.617 [92].

The outbreak indicates how important it is to make prevention (vaccination and treatment) more efficient and faster to prevent the generation of other mutants. The mutations that should be considered in the modified vaccines remain questionable. The main candidates are E484K, N501Y, and L452R in the receptor-binding domain (RBD) (L452R was found in the recently reported Indian variant B.1.617 and the US variant B.1.429 [93]). Based on the SARS-CoV-2 mutation rate, it is expected that an annual revision of the vaccine sequence will be needed for at least the next 2-3 years [94].

As of January 2021, 233 candidates for vaccines against COVID-19 were being developed, with 170 evaluated in the preclinical phase and 63 in the clinical phase [95]. Most vaccine candidates against COVID-19 aim to induce neutralizing antibodies against the Spike protein, preventing the human ACE-2 receptor’s uptake. A striking feature of the COVID-19 vaccine development is the variety of technology platforms, including live attenuated vaccines, viral vector vaccines, recombinant protein vaccines, and DNA and mRNA vaccines [96]. Multiple mutations in the spike protein, including those in the RBD, have led to the globally circulating SARS-CoV-2 strains, referred to as variants of concern, interest, and under monitoring [13].

Besides, the protective effect of non-target vaccines such as the Bacillus Calmette–Guérin (BCG) vaccine is being investigated in randomized, prospective clinical trials in patients infected with SARS-CoV-2. The Murdoch Children’s Research Institute in Australia examines the BCG vaccine, which has been used for nearly a hundred years to prevent tuberculosis by exposing patients to a small dose of live bacteria. Over the years, there has been evidence that this vaccine can boost the immune system and help the body fight other diseases [97]. On November 10, 2020, a study published by Rivas et al. [98] found that among 6201 healthcare workers in Los Angeles, those who previously received the BCG vaccine reported symptoms of COVID-19 less frequently than those who did not receive the vaccination [98]. Strong data about BCG vaccination’s protective role should be concluded before reflecting on practice and vaccination policies.

### 4.2. The Vaccine-Run Winners 

The European Medicines Agency (EMA) issued a conditional marketing authorization on 21 December 2020 of the Comirnaty (BNT162b2) vaccine by BioNTech, in collaboration with Pfizer, on 6 January 2021, the mRNA-1273 vaccine by Moderna Biotech, on 29 January 2021 Vaxzevria (previously AstraZeneca) by AstraZeneca in collaboration with the University of Oxford, and finally on 11 March 2021, the Janssen vaccine by Johnson & Johnson [99]. Companies are testing the vaccine’s efficacy against new virus variants [100]. Notwithstanding, the information released at such an early stage does not answer critical questions about whether the vaccine can prevent the most severe cases or quell the coronavirus pandemic. The question also remains if it can protect people from infection or prevent those showing very mild disease symptoms from spreading the virus. Last but not least, a critical unanswered question is how long the vaccine’s response will last, especially in light of omnipresent reports about the possibility of re-infection [101]. 

On 25 January 2021, Moderna announced that its vaccine effectively protects against the SARS-CoV-2 variants, first identified in the UK and South Africa [102]. Although this evidence has not been fully verified, it comes from testing the blood samples of people who took both vaccine doses. The UK variant had no significant effect on antibody levels. Notwithstanding all those assurances, the company announced the testing of a new vaccine used as a booster against the South African variant [102]. Wu et al. demonstrated reduced but still significant neutralization against the B.1.351 variant following the Moderna vaccine [103].

The University of Oxford, in collaboration with AstraZeneca, has launched a viral vector vaccine. Studies by Madhi et al. [104] show that a two-dose regimen of the AstraZeneca vaccine did not protect against mild-to-moderate COVID-19 due to the B.1.351 variant [104]. Research suggesting that AstraZeneca is less effective against this variant has led to a suspension of its implementation in South Africa [89]. The Russian Direct Investment Fund has also introduced the Sputnik V vaccine with viral vectors. On 2 February 2021, The Lancet published phase III trial results, which concluded that Sputnik V was safe and 91.6% effective in preventing COVID-19. In August 2020, Russia approved the Sputnik V vaccine for the widespread use and announced it as the first registered COVID-19 vaccine on the market [105].

Johnson & Johnson developed a viral vector single-dose vaccine. On 29 January 2021, the company announced phase III of the ENSEMBLE study, which showed that the vaccine is 66% effective in preventing COVID-19. The vaccine was 72% effective in the US studies, but its efficacy dropped to 57% in South Africa, suggesting its lower efficacy against this variant. The J&J vaccine has become the fourth approved by the EMA and granted for emergency use only by the WHO on 12 March 2021 [106].

### 4.3. Further Perspective and Another Challenge

Virus mutants show changes in spike protein-coding regions that are recognized by “neutralizing” antibodies. It raises the possibility that specific mutations may affect the ability of the antibodies to bind the RBD and the N-terminal domain of the target protein [107]. Significantly, in preliminary studies from January 2021, the team of Xie et al. [108] found differences in the affinity of the antibodies produced by the 20 participants against the viruses carrying the N501Y mutation compared to the antibodies produced to combat viruses lacking this change [108]. Evidence is also emerging that the E484K mutation may allow the virus to evade the immune response. In a study published on 28 December 2020, Andreano et al. [109] grew SARS-CoV-2 in the presence of low levels of convalescence serum. Within 90 days, the virus "produced" three mutations that made it resistant to human serum: One was the E484K mutation found in the South African variant; the others are changes in the N-terminal domain found in both the South African and British variants. This suggested that the entire anti-SARS-CoV-2 antibody response was directed against a small fraction of the spike protein [109]. Cohort studies by Challen et al. [110] suggest that variant B.1.1.7 may be associated with increased mortality. Sequencing of the SARS-CoV-2 genome shows a nucleotide substitution rate similar to that of the Ebola virus. All the currently feared variants have many differences from the original Wuhan, and mutations in the spike may alter the interaction with the ACE-2 receptor [110]. 

Thus, it may turn out that the vaccine technology will somehow not follow the new virus outbreaks. Consequently, identifying novel strategies and markers or diagnostic methods in the anti-SARS-CoV-2 strategy will be highly required. Therefore, so far, one of the critical factors in successful COVID-19 elimination is SARS-CoV-2 infection prevention. Importantly, knowing the metabolism of human cells and metabolism of the virus and other similar ones, we hope to adapt antiviral strategies that already exist. 

Thus, we believe that RNA-based vaccines will appear efficient enough to provoke a stable response and long-lasting resistance to SARS-CoV-2 infection. For decades, such vaccines have been studied for the flu, Zika, rabies, and cytomegalovirus (CMV) [111]. Even if vaccines based on RNA or DNA have not resulted in a successful vaccine other than against COVID-19, it is of high importance and interest to proceed with this promising research, especially in the context of sensitivity of the lipid nanoparticles to thermal conditions and transport issues [112]. Effective strategies have already been developed to improve miRNA’s safety, stability, and potential use as vaccine candidates [113]. miRNAs have already been used to construct live attenuated vaccines (LAVs) such as poliovirus LAV, which have miRNA binding sites for let-7a or miR-124a. In addition, miRNA-containing vaccines will not replicate in neuronal cells carrying the relevant miRNA. However, it may still replicate in the gastrointestinal tract and therefore remain to act as immunogens. Thus, miRNAs are also attractive candidates for the development of vaccines and antivirals [114].

## 5. Alternative Approach—miRNA Profiling in Viral Infections

### 5.1. miRNA in Diagnostics and Therapy

miRNA studies began in 1993 since Lee et al. [115] observed their involvement in regulating the key protein, LIN-14, in *Caenorhabditis elegans*. Since then, much research into miRNAs has been made and showed their contribution to various physiological and pathological processes in animals and humans. MicroRNAs (miRNAs/miRs) are highly conserved, small (about 22 nt long), non-coding single-stranded ribonucleic acids (RNAs) that control the expression of complementary messenger RNAs (mRNA). They repress the translation of target mRNA by binding to the 3’ ends in the untranslated regions (UTRs) or to a specific region in open reading frames (ORFs) of an mRNA transcript. miRNAs are named with the miR prefix followed by an identification number. If there are notably the same sequences, extra suffixes (letters or numbers) are provided. The use of -3p and -5p suffixes indicate the 3’ or 5’ end of the miRNA precursor [116]. They are expressed in the nucleus as primary miRNA transcripts (about 1000 nt), processed by the dsRNA-specific endonuclease Drosha into precursor miRNAs (pre-miRNAs). pre-miRNAs (approximately 70 nt) are transported to the cytoplasm and processed by the Dicer into mature miRNAs. A single-stranded, mature miRNA is fused with the RNA-induced silencing complex (RISC), which binds to the untranslated region (3’-UTR) of target mRNA [117] (Figure 3). One miRNA may regulate many mRNAs, but also, one mRNA may be regulated by several miRNAs (full complementarity is not critical). They exert direct impacts by blocking the translation or inducing mRNA degradation. Consequently, by manipulating protein levels, they regulate many cell processes, including cell differentiation, development, growth (also cancer growth), apoptosis, and neurological disorders [118]. miRNAs can also affect the virus replication and respective disease development via direct binding to the viral RNA [119]. Undoubtedly, understanding the role of miRNA in viral infection response may contribute to the identification of novel diagnostic methods and antiviral strategies. 

For instance, one of the hypothesized novel COVID-19 treatment approaches includes designing a mix of miRNAs targeting several ORF and 3’UTR regions of SARS-CoV-2 and a suitable liposomes-like exosome delivery system targeting respiratory tissues [120]. Studies report the differential expression and biological function of miRNAs in airway cells [121], which are the target of SARS-CoV-2. miRNAs may be responsible for the function of dendritic cells, epithelial cells, monocytes, granulocytes, NK cells, and macrophages that participate in innate immune responses [122]. To highlight this, there is still no available drug that increases or inhibits any miRNAs in viral respiratory infections [123]. Nevertheless, for example, MRX34 is the primary artificial miRNA used for advanced hepatocellular carcinoma treatment [124]. Moreover, synthetic miRNAs have been produced to transfect mononuclear cells of peripheral blood with liposomal transporters [125]. 

Thus, not without reason, miRNAs are proposed as promising biomarkers, new targets, and tools in therapeutic approaches [126], but also as prognostic factors in SARS-CoV-2 infection (Figure 3).

Canonical miRNA biogenesis begins with RNA polymerase transcribing a pri-miRNA transcript (>70 nt) from miRNA genes. It is further metabolized by a microprocessor complex (Drosha and DGCR8) to form a pre-miRNA (~65 nt). The pre-miRNA is exported from the nucleus by exportin-5 in a Ran-GTP-dependent manner. In the cytoplasm, it is transformed to a dsRNA of ~22 nt by a complex of Dicer and a dsRNA binding protein (TRBP or PACT) [127]. This dsRNA is then incorporated into the pre-RISC, where strand selection occurs: One strand is degraded, and the other strand remains incorporated into the RISC. By base pairing, miRNAs can form a dsRNA with cellular mRNA, usually in the 3’ region of the UTR, and can either inhibit its translation or induce its degradation [128,129]. In the case of infection, the non-coding SARS-CoV-2 miRNA may interfere with normal cellular homeostasis by upregulating certain host mRNA levels normally controlled by the host miRNA. Consequently, by down-regulating specific host miRNAs, the virus enhances its own replication cycle and attenuates the host’s immune responses [130]. Bioinformatics analysis of the SARS-CoV-2 genome reveals potential miRNAs’ binding sites in various genome regions, e.g., in the critical 5’ UTR regions of ACE-2 or TMPRSS2 [131]. In vitro and in vivo studies also describe various miRNA types with diverse expression trends defined as biomarkers of COVID-19 progression, identified as potential contributors of pathogenesis, possibly serving as biomarkers for severe COVID-19 and potential therapeutic targets [132], all of which are discussed in the article. Arrows indicate up- and downregulation of selected differentially expressed miRNAs in Calu3 cells infected with SARS-CoV-2 or mock from GSE148729 based on studies by Tak-Sum Chow et al. [133]).

### 5.2. The Contribution of miRNAs to the Virus Infection Course

The role of miRNAs has been highlighted in host–pathogen interactions [127,134]. Host’s (human’s) miRNAs can play an antiviral role based on sequence complementarity leading to binding and directing viral RNAs to degradation. Consequently, they can modulate viral infection by controlling the amount of host proteins that affect viral replication [134]. On the other hand, viral miRNAs can also affect the expression of host mRNAs involved in cell proliferation and survival, stress responses, and antiviral responses such as Toll-like receptors (TLRs) or cytokines such as type I interferons. Consequently, the host antiviral immune responses are attenuated. Alternatively, viral miRNAs can reduce virus replication in infected cells, allowing the host cells to survive, go into a latency state, and thereby increase viral spread to other people in the population [135].

Upon entry into the host cell, SARS-CoV-2 releases its genomic RNA. Some viral miRNAs target specific host mRNAs and miRNAs, thereby altering gene expression and modulating the pathways associated with the immune system. At the same time, host cells may change their own miRNA expression profile to defend themselves against the disease [136]. 

It is unknown what miRNAs from the host or the virus mediate the interplay between SARS-CoV-2 and human tissues or affect human immunity [137]. However, the existing relationship between miRNAs and viral infection suggests their therapeutic and/or diagnostic potential in viral diseases. Trobaugh and Klimstra [138] described various interactions between host miRNA and viral RNA (Figure 4), including miRNA–RNA interactions, miRNA–viral genome interactions, miRNA-mediated stabilization of viral RNA genomes, and the modulation of host miRNA levels during viral infection. Changes in the expression of host proteins mediated by miRNAs alter responses to infection, promote viral replication, and maintain miRNA binding sites in the viral genome. Therefore, miRNAs have been identified as promising biomarkers and new targets in therapeutic approaches [138]. 

Due to a rapid mutation rate and diverse virus variants in different world regions, other mechanisms contribute to the interplay, different morbidity/resistance, disease symptoms severity, and mortality rate [139]. Additionally, it must be considered that patients not showing any specific symptoms are not identified (asymptomatic persons account for approximately 40–45% of infections, and they can transmit the virus for an extended period, even longer than 14 days [140]). Consequently, solid verification of the assessment of the role of miRNA in disease development will take some time. 

The variability and distribution of different genotypes in populations is a fact (mainly provided by laboratories through the GISAID initiative [141]). Therefore, it may be that the main challenge is to perform a comparative analysis between patients concerning both host and viral miRNA profile [142] and COVID-19 statistics in different regions, but also a different course of the disease. Thus, it seems critical to evaluate the miRNA profile in a group of patients, control subjects, and patients who recovered from the infection and try to correlate these data to gain diagnostic or prognostic outcomes based on the mechanistic perspective. 

### 5.3. Promising Fields

According to the online miRNA repository, miRbase (release 22.1 [143]), 2600 miRNAs exist in the human genome. About 2000 miRNAs circulate in human fluids [144], potentially regulating hundreds of different targets, including viral transcripts. Noteworthily, miRNA does not have to be 100% complementary to trigger effects [145], which makes it a perfect tool when different mutated variants of SARS-CoV-2 are considered. Consequently, finding miRNAs that are critical for a virus infection can become overwhelming. There are two main strategies to identify potential candidates. First, we can check if the viral infection affects the miRNA profile of the host cell and if some miRNAs are strongly deregulated. This method’s main limitation is that it cannot be stated whether the observed miRNA deregulation is relevant to the virus or if it is only an indirect effect. A second possible approach is to induce the overexpression or inhibition of individual miRNAs and evaluate the phenotypic alterations in target cells subjected to viral infection [146,147]. This approach’s advantage is that it is objective. However, it also has some limitations, especially when it comes to blocking miRNAs. For example, a particular cell line expresses (at a functional level) only ca 100 different miRNAs at most, which implies that it is easy to miss the effect of inhibiting a miRNA that is not naturally expressed in the cell type used for the screen [134]. 

As therapeutics, miRNAs have been extensively analyzed and documented in the treatment of various types of cancers, heart diseases, asthma, pneumonia, etc. [148]. However, in virology, miRNAs’ landscape as diagnostic and interventional medicine is still an unexplored area [149]. Still, Tang et al. identified four microRNA types with different expression levels that could serve as biomarkers of COVID-19 progression and a few miRs (miR-146a-5p, miR-21-5p, miR -142-3p, and miR-15b-5p), possibly serving as biomarkers of severe COVID-19 [150]. All these data support the hypothesis that the miRNA-based antiviral approach may be beneficial in prevention, pathogenesis, and prognosis assessment, as well as COVID-19 treatment. Consequently, broad-scale profiling of the miRNA in the SARS-CoV-2 target cells and tissues is highly required.

### 5.4. miRNAs as Diagnostic Biomarkers

The main benefits of using miRNAs as biomarkers might be their specificity and early detection possibility. It is crucial for improving patient prognosis and limiting the disease’s spread, because the treatment options become limited as the disease progresses [151]. Standard approaches for miRNAs detection involve northern blotting, microarrays, qPCR, and next-generation sequencing (NGS). New miRNA biomarker studies tend to use qPCR and NGS, with Northern blotting and microarrays falling out of favor due to defects, including low sensitivity or specificity and higher total RNA input. Lately, researchers have utilized mass spectrometry to detect miRs [152].

Regarding the sensitivity of miRNA detection methods, for example, serum levels of miR-21, miR-122, and miR-223 have been found to distinguish HIV-positive from HIV-negative patients. In addition, the amount of miR-3162-3p in the plasma of HIV-positive patients has been shown to differentiate new (<one year post-infection) from old (>one year) infections [153]. Interestingly, a study by Yahyaei et al. found that miR-223 was elevated in people who had been repeatedly exposed to HIV but were not infected [154]. A panel of four miRNAs (miR-16-5p, miR-20b-5p, miR-195-5p and miR-223-3p) were developed and tested for their suitability for HIV diagnostics [155]. It highlights miRNAs’ use as disease biomarkers during the infectious period, where conventional markers (antibodies, viral RNA/protein) remain undetectable. Similarly, miRNA profiling in the serum of children with enteroviral infection identified miRNAs (miR-148a, miR-143, miR-324-3p, miR-628-3p, miR-140-5p, and miR-362-3p) that were able to differentiate between infected and healthy patients with a sensitivity of 97.1% and a specificity of 92.7% [156]. These outcomes support the hypothesis that miRNA biomarkers identify infected patients and distinguish different causative factors.

The first symptoms of most viral infectious diseases are often very nonspecific (like fever, pain, headache), providing little or no information about the causative pathogen [157]. In contrast, HSV-1 derived miRNAs show some specificity during active infection (miR-H1) [158] and latency (miR-H2-6) [159] with a specific role in the host response impairment. Thus, it seems that miRNAs detection can be beneficial, especially when the routine diagnostic methods fail. Similar to many aspects of medicine, this approach first attracted the attention of oncologists [160] but is now spreading to other areas, including viral infections and diseases. Again, frontotemporal dementia patients also show characteristic patterns of miRNA expression. Moreover, some distinctive miRNA profiles were brain regions specific to pathology sites [161,162]. 

To date, several registered clinical trials in the miRNA biomarker database have been completed; this includes phase IV studies that monitored selected miRNAs as biomarkers of disease progression in patients receiving FDA-approved drugs. The studies looked at patients to examine these ncRNA transcripts’ profiles in conditions such as diabetes, coronary artery disease, breast cancer, lupus, epilepsy, depressive disorders, stroke, Addison’s disease, influenza, liver disease, and even toxic exposure to factors such as acetaminophen [163]. Additionally, because miRNAs are widely expressed throughout the body, they can be easily measured from peripheral blood, tissue biopsy, urine, saliva, cerebrospinal fluid, and other biological samples [164]. miRNAs can cross the blood–brain barrier, so they can potentially be quantified in routine blood, serum, or plasma tests as measures for a variety of neurodegenerative and neurodevelopmental disorders. Thus, approximately 90% of extracellular or circulating miRNAs can be found in association with proteins, and the remaining 10% is transported in microbubbles such as exosomes and apoptotic bodies [165]. Both proteins and microbubbles protect the miRNAs they carry against RNase degradation and ensure their high stability in an unfavorable extracellular environment [166].

### 5.5. The Role of miRNAs in Viral Infection and Therapy Perspective 

Since we already have some data concerning miRNA metabolism and efficient detection technology, it seems natural to implement these observations in the SARS-CoV-2/COVID-19 assessment. The area would cover the identification of the mechanisms associated with infection susceptibility, symptoms severity, diagnostics, therapy, or immunological response. Analyzing the potential use of miRNAs as therapeutic targets or strategies, we should first divide miRNAs based on their source: Host or viral miRNAs. Given their role, host miRNAs can be classified as anti- or pro-viral depending on their activity when a specific virus enters the host cell. The latter could enable virus replication and infection through the interaction between the viral and host miRNAs, thus fulfilling a pro-viral function [167,168]. Besides, pro-viral miRNAs may promote viral infections by inhibiting antiviral agents such as interferon (IFN) [169]. In contrast, various host miRNAs can show antiviral potential by influencing viral RNA synthesis, blocking viral replication, suppressing pro-viral proteins, or pushing the virus into a latent phase [170,171,172]. Another indirect role of miRNAs in viral infections was also demonstrated, e.g., the contribution of viral miRNAs to the modulation of various signaling pathways such as Wnt and Type I Interferon signaling pathways, NF-κB, PI3K/Akt, and MAPK pathways, and the Notch signaling pathway [173]. Thus, we believe miRNA-targeting strategies could bring significant therapeutic effects. The most efficient and promising technologies are antagomiRs, antisense anti-miR oligonucleotides (AMOs), miRNA sponges, locked nucleic acid (LNA) anti-miRs, and small molecule inhibitors of miRNAs (SMIRs). Chemically modified antisense oligonucleotides showed the most significant effects [174]. However, we are still far from completely understanding the molecular mechanisms behind the complex interplay between miRNA pathways and viral infections. 

### 5.6. miRNA-Based anti-COVID-19 Strategies

miRNAs derived from SARS-CoV and SARS-CoV-2 viruses can target multiple signaling pathways related to the immune response. Still, only some SARS-CoV-2 miRNAs uniquely target autophagy and IFN-I signaling, which may prolong latency within specific hosts without COVID-19 symptoms [137]. Besides, SARS-CoV-2 may modulate several critical cellular pathways, increasing anomalies in patients with comorbidities such as cardiovascular disease, diabetes, or respiratory complications. It may suggest that miRNAs might be a key epigenetic modulator responsible for seriously ill patients with COVID-19 [175].

#### 5.6.1. Bioinformatics—One More Challenge

Potential miRNAs’ function should be initially assessed bioinformatically and/or in vitro before testing in preclinical animal models. Many independent algorithms predict miRNA binding sites in protein-coding genes and their associated biological networks [176]. One of them is TargetScan, which indicates miRNA gene targets based on regions critical for mRNA binding [177]. Basically, computational programs determine the lowest free energy of binding between two selected RNA sequences. Creating miRNA data repositories can further facilitate the identification and evaluation of diagnostic or therapeutic candidates [178,179].

##### Preliminary in Silico Studies

Sardar et al. [180] applied a network biology approach to elucidate key factors involved in miRNA interactions with the host and viral genes. Based on the analysis of the gene regulatory network, it was found that the five major genes, HMOX1, DNMT1, PLAT, GDF1, and ITGB1, are involved in the induction of interferon IFN-α2b, epigenetic modification, and modulation of antiviral activity, making it possible to identify new targets for potential candidates for antiviral drugs. For example, they might directly target the SARS-CoV-2 ORF3a and ORF8 regions, as highlighted in Figure 5. Further analysis of miRNA expression selected 49 miRNAs expressed in the lung tissues. They have also identified 38 miRNAs targeting 143 host genes, whose expression is responsible for antiviral activity [180]. 

Interestingly, six antiviral miRNAs, miR-1-3p, miR-17-5p, miR-199a-3p, miR-429, miR-15a-5p, and miR-20a-5p, have been reported to play a role in respiratory diseases influenza A and respiratory syncytial virus. They were downregulated in lung tissues during viral infection but were overexpressed in normal lung tissues [125,181]. The HMOX1, DNMT1, PLAT, GDF1, and ITGB1 gene expression products may also directly target the SARS-CoV-2 ORF3a and ORF8 regions. Regulation of those genes’ expression has already been described in modulating the immune responses to other viral infections [182,183,184]. It confirms that despite the high mutation frequency in ORF and S genes, the miRNA targets might be conserved and serve as the host’s natural antiviral defense tool. For example, miR-138-5p, miR-622, miR-761, and miR-A3r each have four targets, and miR-15b-5p, miR-18a-5p, miR-A2r, and miR-B1r each have three targets on the SARS-CoV-2 virus. Interestingly, six of these eight miRNAs were found to target the S protein, while seven had two or more targets in the ORF1ab [180,182].

Using computational methods, Khan et al. [137] explained the interaction between the host miRNA and the miRNAs of SARS-CoV and SARS-CoV-2. They also evaluated how these infections differed in the context of miRNA-mediated interactions with the host. As a result, they identified several putative host antiviral miRNAs that could target SARS viruses. Similarly, they managed to predict miRNAs encoded by SARS viruses targeting the host’s genes. Interestingly, it was revealed that several miRNA clusters were associated with increased mortality rates of individual SARS-CoV-2 genomes. In addition, they found miRNAs with experimentally validated antiviral roles; miR-323a-5p and miR-654-5p (predicted for SARS-CoV) were shown to inhibit viral replication in H1N1 Influenza virus infection [185], while miR-17-5p and miR-20b-5p (indicated for SARS-CoV-2) were found to be upregulated in H7N9 Influenza virus infection [186].

Balmeh et al. [187] identified miR-1307-3p (out of 1872 evaluated microRNAs) as the miRNA with the highest affinity for the SARS-CoV-2 genome and infection-associated cellular signaling pathways. They showed that miRNAs played a significant role in the PI3K/Act pathway, endocytosis, and type 2 diabetes and might play a key role in viral entry, proliferation, and development. They also collected over a thousand herbal compounds and performed receptor docking with ACE-2 and TMPRSS2 receptors. Among them, they selected three popular compounds, including berberine, hypericin, and hesperidin, which, in silico, were effective in preventing COVID-19 infection [187]. Fulzele et al. [188] performed an in silico analysis of human miRNAs and found 848 common miRNAs targeting the SARS genome and 873 common miRNAs targeting the SARS-CoV-2 genome. In addition, 315 miRNAs were unique for the SARS-CoV-2, and 290 miRNAs were unique for SARS. They also noted that of the 29 SARS-CoV-2 isolates, 19 had identical miRNA targets. According to his research, the miRNAs with the highest probability of targeting SARS-CoV-2 regions are miR-15a-5p, miR-15b-5p, miR-30b-5p, miR-409-3p, miR-505-3p, and miR-548d-3p [188].

Ivashchenko et al. [189] investigated how miRNAs could protect humans from COVID-19. They found that out of 2565 analyzed miRNAs, three, including miR-6864-5p, miR-5197-3p, and miR 4778-3p, interacted with SARS-CoV SARS-CoV-2 RNAs [189]. They also identified cc-miR (completely complementary miRNA) in coronavirus RNA. The cc-miRs for SARS-CoV, MERS-CoV, and SARS-CoV-2 had no target genes among the human genes, meaning no side effects of these cc-miRs in human mRNAs translation. cc-miR can show therapeutic potential when incorporated into exosomes or other vesicles and are transported into the blood or lungs by inhalation. Introducing cc-miRs into the bloodstream might stop the virus from reproducing in the blood and organs. This method could be applied to other viruses as well [189]. 

Hosseini et al. [190] discovered miR-574-5p, miR-214, miR-17, miR-98, miR-223, and miR-148a among human miRNAs capable of binding to SARS-CoV-2 [190]. Computational prediction of miRNAs and their binding affinity to SARS-CoV-2 should be considered very carefully as the results must be verified experimentally [191,192,193]. Thus, Chow et al. [133] predicted in silico 128 human microRNAs in the lung epithelium that could target SARS-CoV-2. Furthermore, the experimental research confirmed that six of those microRNAs are simultaneously expressed in SARS-CoV-2 infection in vitro [133].

#### 5.6.2. In Vitro and In Vivo Analyses

Various cell culture platforms are available to estimate the mechanisms and possible therapeutic efficacy of candidate miRNAs in vitro (Figure 6). Primary cells, immortalized cell lines, and induced pluripotent stem (IPS) cells are readily available for epigenetic manipulation. For example, IPS cells allow the modulation and monitoring of biological pathways along various stem cell lines from the skin tissue [194,195]. High-throughput screening and confirmation of bioinformatics predictions in vitro significantly facilitated and accelerated the preclinical investigations of putative therapeutic transcripts [196].

##### Recent In Vitro and In Vivo Studies

Wyler et al. [197] examined three human cell lines Caco-2, Calu-3, and H1299, for the miRNA profile for both SARS-CoV-1 and SARS-CoV-2 infection. The studies showed potent induction of protective and inflammation-related microRNAs, such as miR-146 and miR-155. Calu-3 infection with SARS-CoV-2 resulted in an approximately 2-fold increase in gene expression of IFN-stimulated response elements and expression of cytokines, such as CXCL10 or IL6 compared to SARS-CoV-1 infected cells. Interestingly, in a study by Wyler et al., lung damage caused by the acute respiratory distress syndrome (ARDS) was reduced by eliminating miR-155, which means that this particular miRNA may be a potential target for the treatment of COVID-19 [197]. 

Xie et al. [198] confirmed that let-7 blocks SARS-CoV-2 replication by targeting S and M proteins. Meanwhile, let-7 inhibits the expression of many inflammatory factors, including IL-1β, IL-6, IL-8, TNF-α, and VEGFα. More importantly, C1632, a small molecule serving as a let-7 stimulator, can upregulate let-7 expression, thereby reducing viral replication and secretion of inflammatory cytokines. Previously, C1632 was shown to have low toxicity to cultured cells and mice. Therefore, the safety and beneficial effect of C1632 in inhibiting SARS-CoV-2 replication and suppressing viral-induced inflammation should be strongly emphasized. Further research into the safety and efficacy of C1632 may help promote its clinical use [198].

#### 5.6.3. Molecular Targeted Therapy—The Progress and Future Strategy Perspective

There are reports of diagnosed post-COVID-19 myocarditis by electron microscopy [207]. Autopsy studies found that 5 out of 12 COVID-19 victims had SARS-CoV-2 mRNA in the myocardium, in whom clinically suspected myocarditis was confirmed by biopsy endomyocardial (EMB) with evidence of persistent SARS-CoV-2 mRNA in the heart. Nasopharyngeal swabs did not confirm SARS-CoV-2, so myocarditis may also be a delayed sequel to asymptomatic or cured COVID-19, not only the acute and often life-threatening myocarditis in active COVID-19 [207]. Thus, there is a need for more advanced diagnostics and clinical trials on disease progression, novel strategies, treatment options, and the long-term prognosis of SARS-CoV-2 infection. 

As discussed above, reports already indicate human miRNAs that potentially target SARS-CoV-2 (i.e., miR-654-5p, miR-198, miR-622, and miR-323a-5p from VIRmiRNAs and miR-17-5p, miR-20b-5p, and miR-323a-5p from host miRNAs) as well as H1N1 or H7N9. Noteworthily, the therapeutic potential of miRNAs was also reported in the hepatitis C virus infection [137]. Other studies documented strong upregulation of the miR-200c-3p/miR-141-3p cluster after the H5N1 avian influenza virus infection. It was also found that miR-200c-3p and miR-141-3p could directly target the 3’ UTR region of ACE-2 since transfection with appropriate miRNA mimics and inhibitors showed a decrease or increase in ACE-expression, respectively [208]. 

Recent studies have also shown that ACE-2 is expressed in cardiomyocytes, and some miRs (e.g., miR-200c) downregulate the levels of the ACE-2 mRNA and the ACE-2 protein in rat primary cardiomyocytes, as well as in human induced pluripotent stem cell-derived cardiomyocytes (hiPSC-CMs) [209] (Figure 6). 

Overexpression of miR-200c suppresses ACE-2 expression in both rat and human cardiomyocytes. Further experiments to investigate the potential of miR-200c to reduce ACE-2-mediated SARS-CoV-2 infection using the hiPSC-CM are warranted. Despite this, careful consideration should be given to the conflicting effects of ACE-2 on cardiovascular disease and COVID-19 infections. ACEI and ARB are widely administered to patients with heart failure, which reduces heart stress and leads to increased ACE-2 expression and activity. However, increased ACE-2 levels may also make cells more susceptible to SARS-CoV-2 infection [210]. Thus, establishing a therapeutic time window for ACE-2 targeting in COVID-19 patients with cardiovascular diseases (CVDs) is a significant challenge. Nevertheless, miR-200c-based therapy [208,209] could help to reduce treatment time by targeting *ACE-2* in COVID-19 patients with CVDs. Thus the concept of targeting viral nucleic acids (both RNA as well as miRNAs) that would attenuate its toxic potential seems promising [211].

#### 5.6.4. Alternative Approach

Several potential therapeutic strategies targeting ACE-2 have already been proposed to combat COVID-19. However, because of the critical metabolic and hemodynamic roles of ACE-2, including the regulation of glucose homeostasis [212] and the cleavage of angiotensin I and angiotensin II [210], these approaches may lead to severe clinical problems [213]. Therefore, transmembrane protease serine 2 (TMPRSS2), another main co-factor needed by the SARS-CoV-2 to enter human cells, might be an alternative target for COVID-19 [214]. This gene encodes a protein belonging to the serine protease family. It comprises a type II transmembrane domain, a class A receptor domain, a cysteine-rich scavenger receptor domain, and a protease domain. It is upregulated by androgenic hormones in prostate cancer cells and downregulated in androgen-independent prostate cancer tissue. It facilitates virus entry into host cells by proteolytic cleavage and activation of viral envelope glycoproteins. Viruses that use this protein to enter cells include the influenza virus and, among others, human coronaviruses HCoV-229E, MERS-CoV, SARS-CoV, and SARS-CoV-2 [215]. The development of new approaches to regulate the expression of ACE-2 and TMPRSS2 is gaining importance in the ongoing pandemic context. Various miRNA isoforms (isomiR) may mediate the regulation of ACE-2 and TMPRSS2 expression. Studies performed by Nersisyan et al. [216] showed that 5B lysine-specific demethylase (JARID1B) could indirectly affect ACE-2/TMPRSS2 overexpression by suppressing the transcription of let-7e/mir-125a and mir-141/miR-200 miRNAs that target these two genes. Apart from the crucial role of both enzymes in the penetration of SARS-CoV/SARS-CoV-2 into the cell, ACE-2 and its miRNA have also been shown to contribute to ARDS development. Earlier studies have already indicated a role for ACE-2 in acute lung injury, as lung ACE2 deficiency exacerbated ARDS’s pathogenesis [217]. Moreover, analysis of single RNA cell sequencing data strongly supports the existence of such interactions in other cells and indicates that most human ACE-2 and TMPRSS2 cells are not expressed without JARID1B. In particular, the high levels of JARID1B, ACE-2, and TMPRSS2 expression in respiratory epithelial cells indicate that further research could improve our understanding of the pathogenesis of viral infections such as COVID-19 [216,217]. A study by Matarese et al. [218] showed for the first time that miR-98-5p directly targeted the 3’ UTR of TMPRSS2. They also provided the first evidence of the actual expression of TMPRSS2 in human endothelial cells (including lung and umbilical vein) (Figure 6). It was previously confirmed that miR-98 reduced endothelial dysfunction by stabilizing the blood–brain barrier, where a significant proportion of the damage is due to inflammation caused by pro-inflammatory factors produced in the brain and the involvement of endothelial leukocytes and by improving neurological outcomes in the case of ischemia or reperfusion [219]. Through bioinformatics analysis, Matarese et al. also identified miR-4500 and miR-4458 [218]. However, no evidence exists in the literature about their endothelial dysfunction roles, while other reports [220,221,222] precisely suggest a role for miR-98-5p in endothelial cells. TMPRSS2 is a highly polymorphic gene, and specific genetic variants of TMPRSS2 have been classified, indicating that their frequencies change with geographic location and origin [223,224,225]. Of course, further studies are needed to determine the effect of miR-98-5p on those variants, as only in vitro experiments were performed to test the relationship between miR-98-5p and TMPRSS2 mRNA without verifying the effect in vivo [218].

Matsuyama et al. demonstrated that TMPRSS2 was expressed in lung tissues, and it could be the primary determinant of viral tropism and pathogenicity at the initial site of COVID-19 [226]. Furthermore, TMPRSS2 might promote viral spread by the diminished viral recognition through neutralizing antibodies [227]. The crucial role of TMPRSS2 in COVID-19 is confirmed by the observation of increased expression in bronchial epithelial cells of male patients compared to female patients [228], which may be the primary explanation for the previously described finding of an independent male gender association with severe COVID-19 [229]. 

Another therapeutic option might be based on intervention in the interactions between ACE-2 and histone modifiers such as HAT1, HDAC2, and JARID1B. Their efficacy in lungs has previously been demonstrated by Pinto et al. [230]. In particular, the interaction between the HDAC2 protein and the major viral protease Nsp5 was shown. As suggested, Nsp5 could inhibit the transport of HDAC2 to the nucleus, thus altering its activity. Interestingly, valproic acid (fatty acid derivative and anticonvulsant, initially used to treat bipolar disorder) has been shown to directly inhibit JARID1B in human embryonic kidney cells (HEK 293) [231]. Repression of the histone modifier could alter the cell’s response to viral infection by involving ACE2 and TMPRSS2 genes [232]. However, additional experiments are necessary to elaborate the network. The latest study by Kaur et al. [233], published in January 2021, indeed confirms that TMPRSS2 is a new molecular target for SARS-CoV-2 early treatment and prevention (Figure 6). They recognized miR-214, miR-98, and miR-32 with the prospective therapeutic potential to silence the TMPRSS2. These three miRNAs showed a strong binding affinity to *TMPRSS2* with the highest probability of interactions, especially miR-32, whose role was already raised in in vitro studies. However, further and detailed in vivo validations are needed for the ultimate mechanistic functionalities and the comprehensive understanding of the COVID-19 course [233].

### 5.7. Sensitivity and Specificity of microRNA Profiling in COVID-19 Patients 

The analysis of the hundreds of thousands of miRNAs associated with COVID-19 has one goal: To find an effective clinical biomarker, promising diagnostic and/or prognostic factor, or therapeutic target and tool. It is crucial and necessary to consider the sensitivity and specificity of miRNA determination in the plasma of COVID-19 patients because of numerous potential candidates.

Donyavi et al. [234] studied the specificity of the miRNA profile that may act as new biomarkers for differentiating acute COVID-19 disease from the healthy controls and those in the post-acute phase of the disease. The expression level of miR-146a-3p, let-7b-3p, miR-29a-3p, and miR-155-5p was assessed in the peripheral blood cells of patients in and after the acute disease phase and healthy groups by qPCR. The miRNAs specificity and sensitivity were checked by receiver operating characteristic (ROC) analysis. The expression of all examined miRNAs in COVID-19 patients turned out to be significantly higher than in the healthy group. Therefore, the expression pattern of the miRs in the post-acute phase was significantly different from the acute disease phase. ROC analysis showed that miR-29a-3p, miR-155-5p, and miR -146a-3p could serve as a new biomarker for diagnosing COVID-19 with high specificity and sensitivity. In addition, miR-29a-3p and miR-146a-3p may act as new biomarkers for distinguishing acute from post-acute COVID-19. Hence, cellular miRNAs may be used as promising biomarkers for diagnosing and monitoring COVID-19 [234].

The profile of circulating miRNAs in hospitalized COVID-19 patients and the evaluation of their potential as biomarkers have also been studied by Gonzalo-Calvo et al. [235]. Plasma miRNA profiling related to disease severity and mortality in ICU patients was performed using RT-qPCR. The predictive models were constructed using the least absolute shrinkage and selection operator (LASSO). The analysis identified miR-148a-3p, miR-451a, and miR-486-5p, which distinguish ICU patients from ward patients. The expression of miR-192-5p and miR-323a-3p differentiated ICU nonsurvivors from survivors. The levels of those miRNAs were correlated with the duration of stay in the ICU. Therefore, plasma miRNAs might be considered new tools to help clinicians predict early deterioration in life among patients [235].

The miRCURY LNA miRNA miRNome qPCR panels were applied to characterize circulating miRNAs in the plasma of COVID-19 patients compared to healthy donors. To validate the results, a qRT-PCR reaction was performed. The ROC curve analysis was used to assess the diagnostic accuracy of the most deregulated miRNAs. miR-17-5p and miR-142-5p were significantly downregulated, while miR-15a-5p, miR-19a-3p, miR-19b-3p, miR-23a-3p, miR-92a-3p, and miR-320a increased. Analyses of the ROC curve and AUC area for miR-19a-3p, miR-19b-3p, and miR-92a-3p indicated a high diagnostic value of these miRNAs, suggesting that they could serve as a potential diagnostic biomarker and/or therapeutic target during SARS-CoV-2 infection [236].

Sabbatinelli et al. [237] examined the serum levels of miRNAs regulating inflammation, miR-21-5p, miR-146a-5p, and miR-126-3p in 29 COVID-19 patients receiving tocilizumab with a sex- and age-matched control group. Among patients who did not respond to the treatment, the worst test results (the highest increase in plasma IL-6 concentration) occurred in those with the lowest serum miR-146a-5p levels. These data show that miR-146a-5p could serve as a biomarker and provide clues about the molecular relationship between inflammation and the clinical course of COVID-19 [237]. They concluded that blood biomarkers, such as miR-146a-5p, might represent a molecular relationship between inflammation and the clinical course of COVID-19 [237]. Sequencing the non-coding RNA and mRNA transcriptomes isolated from red blood cell-depleted whole blood of moderate to severe COVID-19 patients helped define four types of microRNAs with different expression trends that could serve as biomarkers of COVID-19 progression. MiR-146a-5p, miR-21-5p, miR-142-3p, and miR-15b-5p have been identified as potential contributors to disease pathogenesis, possibly serving as biomarkers of severe COVID-19 or as therapeutic targets candidates [237] (Figure 6).

Studies conducted by Li et al. [238] on total RNA extracted and purified from the peripheral blood of ten COVID-19 patients and four healthy donors found that compared to healthy controls, 35 miRNAs were elevated and 38 miRNAs were lowered in people with COVID-19. The most elevated and reduced genes are presented in Table 1. Interestingly, miR-16-2-3p was the most upregulated miRNA, with a 1.6-fold change from the control. Moreover, miR-6501-5p and miR-618 expression were 1.5 times higher in COVID-19 patients than in healthy donors. Meanwhile, miR-627-5p was reduced the most, with a 2.3-fold change from the control. Thus, it is postulated that the differential expression of miRNAs in COVID-19 patients might regulate the immune response and viral replication during viral infection [238].

Recently, circulating miRs (miR-21, miR-126, miR-155, miR-208a, and miR-499) were analyzed in a cohort study of mechanically ventilated COVID-19 patients (*n* = 18) and healthy subjects (*n* = 15) [239] (Figure 6). The serum concentration of miR-21, miR-155, miR-208a, and miR-499 was significantly increased in COVID-19 patients compared to healthy controls. The inflammation and cardiac myocyte-specific miRs here were upregulated in critically ill COVID-19 patients. miR profiles also differentiated patients between severely ill, mechanically ventilated with influenza or COVID-19, indicating a specific response and a heart role in COVID-19. The upregulation of miR-21, miR-155, miR-208a, and miR-499 in COVID-19 survivors might be a predictor of chronic myocardial damage and inflammation. Of note, despite troponin levels being higher in the influenza-ARDS group, myocardial-specific miR-208a and miR-499 were more upregulated in COVID-19 patients [239].

The team of Farr et al. [240] profiled circulating miRNAs in ten COVID-19 patients. They observed 55 miRNAs whose expression was changed at the early stage of the disease. MiR-31-5p showed the highest overexpression. MiR-423-5p, miR-23a-3p, and miR-195-5p also showed altered expression [240] (Figure 6). 

Centra el at. [241] examined the lungs of patients with severe airway injuries and thrombotic events who died due to COVID-19. They identified miR-26a-5p, miR-29b-3p, and miR-34a-5p as regulators of target mRNAs involved in endothelial and inflammatory signaling pathways as well as viral diseases. They revealed the tight interconnection of miRNAs with endothelial activation/dysfunction. Reduced expression levels of selected miRNAs were observed in lung biopsies of COVID-19 patients compared to controls. Their findings indicate a significant contribution of miR-26a-5p, miR-29b-3p, and miR-34a-5p in endothelial dysfunction and inflammatory response in patients with severe lung injury and thrombosis [241].

From patients’ transcriptomic data, McDonald et al. [242] identified a circulating miR-2392 directly involved in the SARS-CoV-2 mechanism during host infection. miR-2392 has been necessary to increase inflammation, glycolysis, and hypoxia and promote many of the symptoms associated with COVID-19. miR-2392 was present in the blood and urine of the COVID-19 patients but was not detected in COVID-19 negative patients. They also developed a novel miRNA-based antiviral therapeutic agent targeting miR-2392 in in vitro and hamster in vivo models, which significantly reduces SARS-CoV-2 viability and might have the potential to inhibit COVID-19 in the host [242].

Papannarao et al. [243] determined that early changes in the miRNA profile may be the primary molecular mechanism increasing the risk of COVID-19 infection in obese individuals. Real-time PCR of plasma samples for circulating miRNAs showed a significant upregulation of miR-200c and a slight increase in miR-let-7b in obese patients. In addition, it was associated with a significant reduction in angiotensin 2 converting enzyme activity. Thus, they showed that increasing miR-200c levels might increase the susceptibility of obese people to COVID-19, and circulating miR-200c may be a potential biomarker in the early identification of people at risk of severe COVID-19 [243] (Figure 6).

All the above-mentioned studies confirmed that SARS-CoV-2 infection elicited a strong response in the host’s miRNAs profile and might improve COVID-19 detection. It also indicates possible strategies for the development of antiviral drugs against SARS-CoV-2 through miRNA regulation. 

## 6. Alternative miRNAs-Associated Pathways

### 6.1. The DEAD-Box Helicases: The Double-Edged Sword in Viral Infections

The host enzymes that play a key role in the virus replication are DEAD-box helicases, although the exact mechanism of their interference with replication is unknown. DEAD-box helicases have been described as ATP-dependent chaperones that reconfigure RNA by disrupting secondary and tertiary RNA–RNA or RNA–protein interactions [244,245,246]. They were named because of the standard DEAD region (Asp-Glu-Ala-Asp). Thirty-seven DEAD-box helicases have been described to influence transcription, splicing, miRNA biosynthesis, translation, and RNA degradation in humans. They may also be involved in cell–virus interactions. Whether they support viral replication or activate the host’s immune response remains a mystery. However, it seems that viruses use DEAD-box proteins for many non-classical functions to complete the replication cycle. Numerous helicases, including DDX1, DDX3, DDX5, and DDX17, are involved in viral survival and maintenance [247,248,249].

Despite the proven roles of DEAD-box helicases and their interaction with RdRp proteins, analogous studies on SARS-CoV-2 have not yet been described. Such work would enable further structural and functional comparisons of host DEAD-box RNA helicases and suggest an intervention at the level of miRNAs biogenesis. In addition, the study of their diverse functional roles in viral replication can be used to propose alternative therapeutic strategies [250,251].

### 6.2. Possible Interference with the miRNA Biogenesis Pathway

The canonical maturation of miRNAs includes the primary miRNA transcript (pri-miRNA) production by the RNA polymerase II/III and cleavage of the pri-miRNA by the Drosha–DGCR8 complex in the nucleus. The resulting precursor, pre-miRNA, is then exported by Exportin-5–Ran-GTP from the nucleus. The RNase Dicer in complex with the double-stranded RNA-binding protein TRBP cleaves the pre-miRNA to its mature length in the cytoplasm [252,253]. Increased susceptibility to vesicular stomatitis virus infection has been reported in mice with the Dicer mutation [254]. Analysis of IFN-stimulated genes and miRNA expression associated with inflammation and immunity in Dicer-deficient mice confirmed lower survival in these animals, while reduced viral miRNA maturation could reduce the pathogenicity [255]. On the other hand, mice with the Dicer mutation also showed decreased cellular miRNA synthesis, which lowers the inflammatory response following murine cytomegalovirus infection, increasing the host’s compliance [255,256,257,258]. Therefore, it is not easy to evaluate the exact contribution of pathogenic viral miRNAs to protective cellular miRNAs in a complex in vivo model. However, this observation could have significant repercussions on human health, where varying DICER expression levels may account for differences in susceptibility to viral infections, possibly including SARS-CoV-2 [134].

### 6.3. The miRNAs Delivery Systems

AntagomiRs are a new class of oligonucleotides that silences endogenous miRNAs and enables a novel therapeutic approach with much greater adaptability and a broader range to the activity’s purpose. Such therapies can work with good effects in people suffering, for example, from severe conditions caused by cytokine storm syndrome. AntagomiRs, delivered by systemic administration of polymer-based nanoparticles surrounding oligonucleotides [146,259], can be used to precisely adjust the activity of crucial miRNAs involved in the inflammatory process. Thus, they may be efficient in COVID-19 patients with severe conditions due to the CSS. Antagomirs are easy and cheap to produce, mainly when made on a large scale. It could be customized upon the immunological hallmarks of a given condition to block the cytokine storm and prevent patients from having severe, life-threatening complications associated with COVID-19. Taken together, antagomirs can be used to precisely tailor the action of a key miRNA that contributes to the inflammatory process in COVID-19 patients and improves these patients’ clinical symptoms [260]. Though the delivery of miRNAs and their antagomirs to target sites remains a challenge due to low cellular uptake and nuclease degradation, various delivery systems are still being tested. In particular, polymer-based non-viral vehicles have demonstrated advantages such as versatile structural modifications and protection of unstable miRNAs [261]. Poly(lactic and co-glycolic acid), chitosan, polyethyleneamine, and polyamidoamine dendrimers may also be carriers for efficient miRNA or anti-miRNA delivery systems. However, in most clinical trials, chemically modified miRNAs are applied without a specific delivery system. Importantly, the main problems associated with miRNA therapy are low circulation/half-life and toxicity of the delivery vehicles. Formulation modifications may improve miRNA stability but may also not result in some incorrect translation [262]. On the other hand, various viral vectors (adenoviral, retroviral, and lentiviral vectors) are widely used for preclinical and clinical purposes. However, their poor miRNA loading efficiency, off-target toxicity, and immunogenicity make their use considerably tricky. Therefore, the development of non-viral microRNA delivery vectors is also essential [263]. miRNAs targeting oncogenes such as microRNA-7 (miR-7) are delivered into the body and target cells using lipid nanoparticle formulations [264]. Nanotherapeutic agents seem to be advantageous for efficient miRNA delivery to host cells due to their small size and low molecular weight. Nanoparticle cores protect miRNAs against degradation and ensure better circulation in the body. Due to the size range of the nanocarriers, targeted delivery of miRNAs is achieved [264]. The use of miRNAs with nanoparticle-based delivery has great potential and could become a novel therapeutic approach to combat the COVID-19 pandemic. They are mainly inorganic nanoparticles, polymer particles, and lipid nanoparticles [265]. They can also be polymer-based delivery systems such as PEI or PLGA. Lipid-based (complexed or encapsulated miRNAs within the membrane-like surface of the lipoplex/liposome) are also a widely used category of carriers [266]. Preparations such as lipofectamine, SiPORT, and DharmaFECT are already used in vitro and in vivo [267,268]. Another approach is based on exosome-like vesicles derived from plants known as edible nanoparticles (ENPs) [269]. They can be filled with microRNAs in a bioavailable form. Interestingly, regulation of human transcripts by plant miRNAs has recently been demonstrated. ENP was isolated from ginger and grapefruit plants, and qRT-PCR miRNA expression studies selected a total of 260 miRNAs derived from ENP. Twenty-two miRNAs were identified that could potentially target the SARS-CoV-2 genome. Interestingly, miR-530b-5p specifically targeted the ribosome slippage between ORF1a and ORF1b. Since the administration of ENPs leads to their accumulation in lung tissues in vivo, ENP-derived miRNAs targeting the SARS-CoV-2 genome have the potential to be developed as an alternative therapy [270].

## 7. Conclusions

The COVID-19 morbidity risk and resistance status are supposedly associated with ACE-2 levels and the virus variants, tuberculosis vaccination (immune system status) or vitamin D intake, and other molecular markers. Unfortunately, no association has been firmly established so far. It seems that the metabolic state and general condition or coexisting diseases and some endogenic mechanisms and genetic variants play an essential role in COVID-19 morbidity. SARS-CoV-2 can modulate critical cellular pathways that might provoke severe symptoms in patients with cardiovascular diseases, diabetes, breathing complications, etc. Specifically, the severity of the symptoms was associated with some epigenetic factors, i.e., miRNA profile. As suggested, miRNAs of both host and SARS-CoV-2 could play a role in the pathogenesis that implies potential direction in the prediction, diagnostics, or even therapeutic strategy. All the premises are based on previous experience with miRNA’s role as an antiviral tool or inducer of the innate and adaptive immune system. Additionally, several DNA and RNA viruses were reported to produce miRNAs known as viral miRNAs (v-miRNAs) to trick the host immune response [271]. Besides, many human miRNAs seem to target viral genes and their functions, including interfering with replication, expression, and translation. The dual functions of host miRNAs cannot be underestimated in viral replication modulation. Host miRNAs may show an antiviral role useful for the host or might promote viral replication and infection by miRNA–viral genome interaction. The prospective investigations on the role of miRNAs in respiratory viral infections can open up a new horizon for the application of miRNAs to prevent and diagnose viral infections. 

Due to the lack of effective treatment strategies for COVID-19, innovative approaches must be sought. A detailed understanding of the molecular mechanism of SARS-CoV-2 pathogenesis remains elusive. There are, of course, already described strategies to tackle the SARS-CoV-2, including inhibiting virus multiplication, inhibiting virus entry through receptors (blocking receptors), blocking viral proteins, but also searching for an interplay between SARS-CoV-2 and host’s miRNAs, an epigenetic regulator, implicating disease complexity. The Human Genome Project’s pioneering discoveries are changing clinical research strategies, with new miRNA therapeutics emerging, albeit remaining elusive to current treatment options. The development of bioinformatics programs to identify miRNA binding sites in target genes and their corresponding biological pathways, together with the expanding platform of preclinical in vitro and in vivo research models, has helped accelerate miRNA translation into clinical medicine. It may be that the expression levels of mentioned miRNAs may offer promising diagnostic value and severity prediction of the COVID-19 symptoms. Furthermore, since miRNA as a tool might modulate the infection step and viral replication and translation, it must be considered a potent therapy element. In virology, the landscape of miRNAs as diagnostic and interventional medicine is still an unexplored area of study, so the challenge is to elucidate the expression of specific miRNAs in COVID-19 implementation as an effective treatment strategy. 

## Figures and Tables

**Figure 1 ijms-22-08663-f001:**
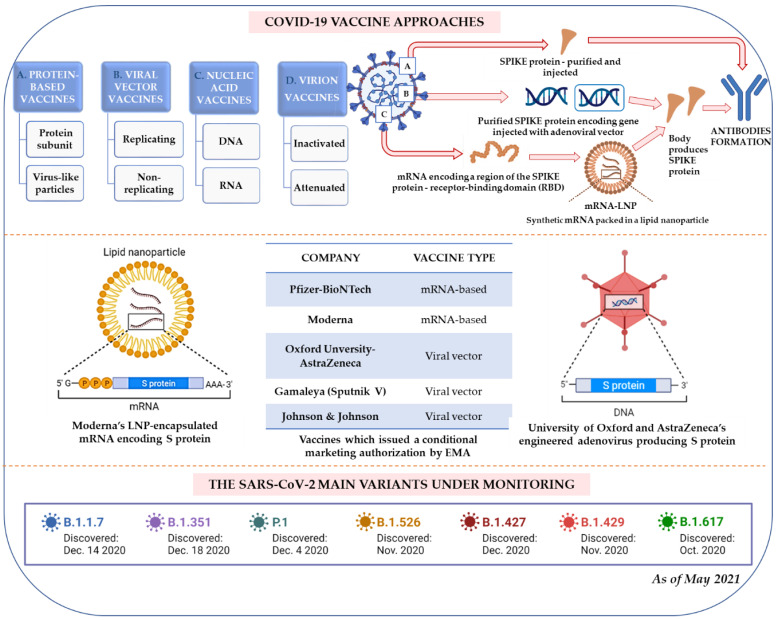
SARS-CoV-2 vaccine approaches and main SARS-CoV-2 variants currently under monitoring.

**Figure 2 ijms-22-08663-f002:**
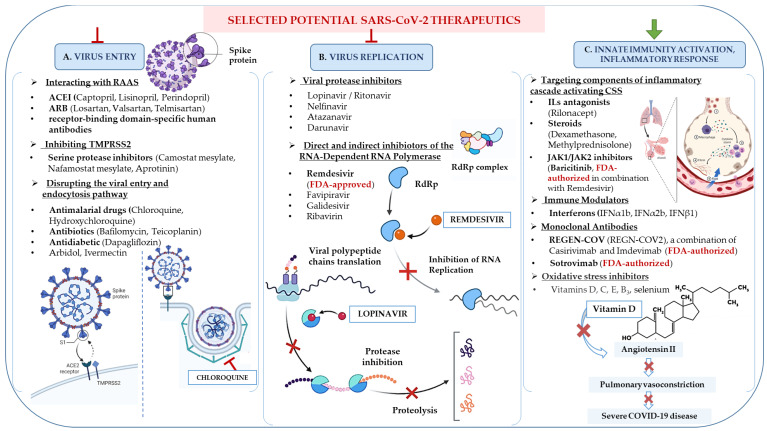
Potential mechanisms of SARS-CoV-2 therapeutic approaches.

**Figure 3 ijms-22-08663-f003:**
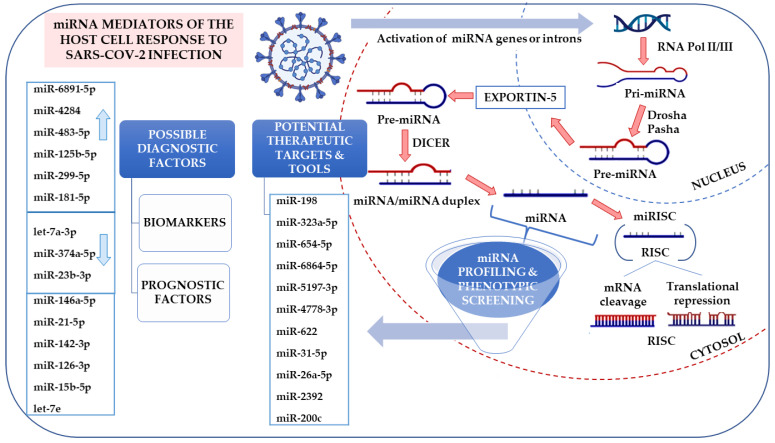
Schematic presentation of host’s miRNA formation mechanism, miRNA-mediated cellular regulation, and possible benefits of miRNA profiling in COVID-19-patients.

**Figure 4 ijms-22-08663-f004:**
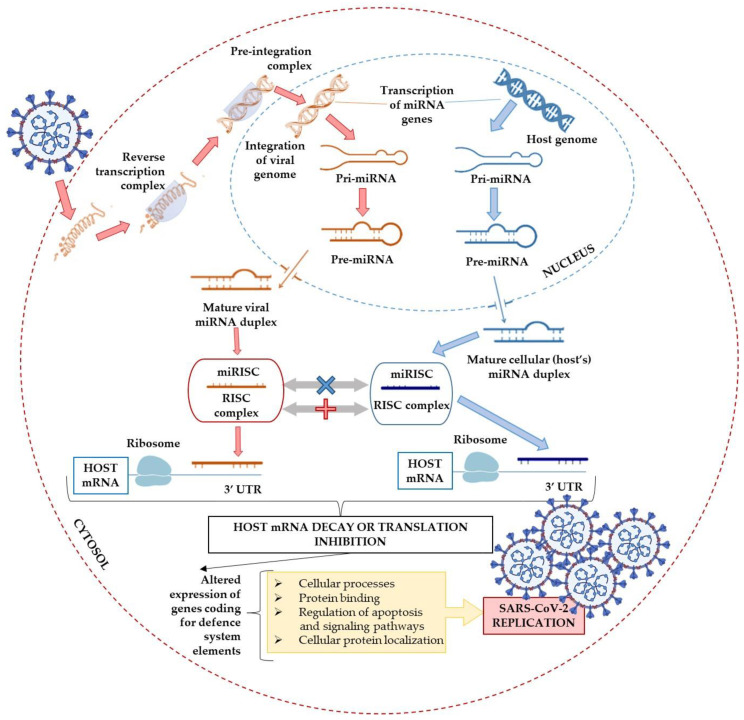
Possible mechanism of viral genome integration into the host’s cell and contribution of the SARS-CoV-2 miRNAs to the infection efficacy.

**Figure 5 ijms-22-08663-f005:**
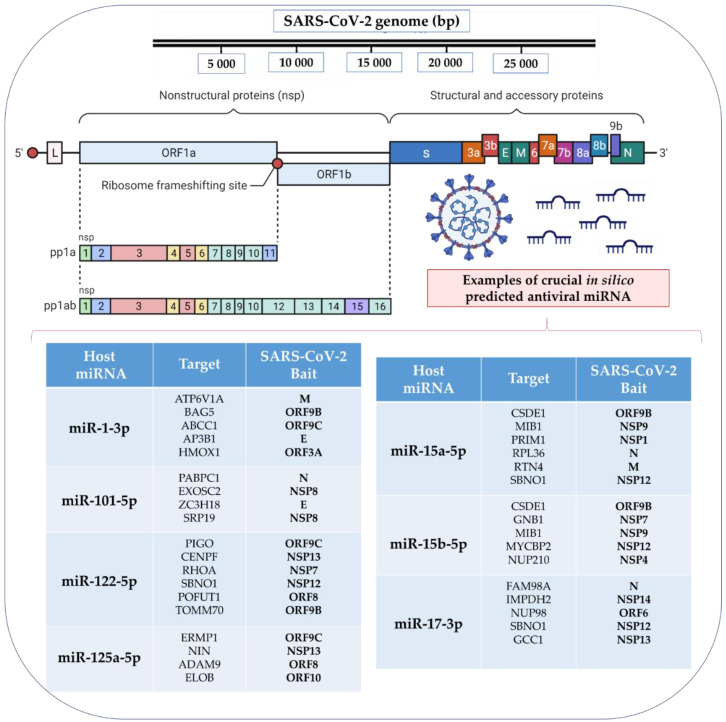
Schematic key elements of the SARS-CoV-2 genome and a reference to the in silico analysis by Sardar et al. [180] of host’s miRNAs with antiviral activity, their targets, and virus baits. Among others presented, five major genes (HMOX1, DNMT1, PLAT, GDF1, and ITGB1) directly targeting the SARS-CoV-2 ORF3a and ORF8 proteins were already described to play a role in modulating antiviral immunity and epigenetics [182,183,184].

**Figure 6 ijms-22-08663-f006:**
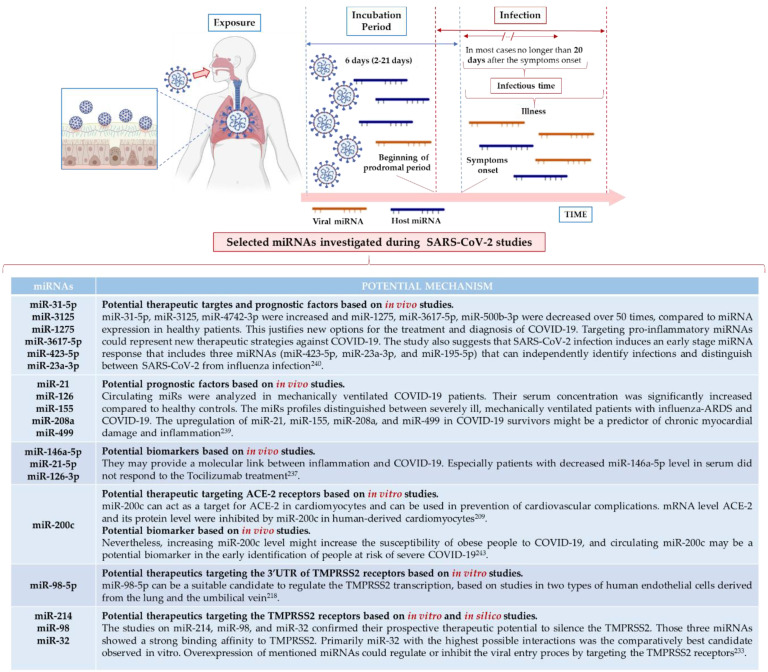
microRNAs alterations during SARS-CoV-2 infection in in silico, in vitro, and in vivo studies and their potential mechanisms of action. After exposure, the viral infection usually goes through three stages: (1) The incubation period, in which the pathogen is present and multiplies, but the patient shows no symptoms until the onset of the symptoms, and the infectious period starts before the symptom onset and might be extended in patients with chronic comorbidities; (2) the prodromal period, occurring after the incubation period, in which the pathogen continues to multiply and cause not-specific, too-general symptoms to indicate a disease, such as pain, fever, soreness, and swelling, typically resulting from the immune response; and (3) the actual period of illness, where the symptoms are most severe and obvious [199,200]. Most adults are likely to remain contagious up to 20 days after the onset of symptoms. However, there have been reports of people shedding a replication-competent virus for more than 20 days due to severe immunosuppression [201,202,203,204,205]. Thus, the released miRNAs of both the host and viral miRNAs can reflect the disease stage and provide important clinical information, especially when the causative pathogen is difficult to detect. SARS-CoV-2 can cause asymptomatic, mild, and severe diseases. Circulating miRNAs might differentiate these stages and their final prognosis [206].

**Table 1 ijms-22-08663-t001:** The most up- and downregulated miRNAs from peripheral blood of COVID-19 patients in comparison to the healthy donors’ samples. Expression levels of various miRNAs were detected by sequencing, and correlation analysis was performed on the target genes. The results confirm that miRNAs in COVID-19 patients could regulate immune responses and viral replication during viral infection [238].

Downregulated miRNAs	Upregulated miRNAs
miR-183-5p	miR-16-2-3p
miR-941	miR-5695
miR-627-5p	miR-618
miR-144-3p	miR-10399-3p
miR-21-5p	miR-6501-5p
miR-20a-5p	miR-361-3p
miR-146b-5p	miR-4659a-3p
miR-454-3p	miR-142-5p
miR-18a-5p	miR-4685-3p
miR-340-5p	miR-454-5p
miR-17-5p	miR-30c-5p

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
