# Peer review of "Anti-SARS-CoV-2 Strategies and the Potential Role of miRNA in the Assessment of COVID-19 Morbidity, Recurrence, and Therapy"

_ijms, 2021, doi:10.3390/ijms22168663_

Round 1

Reviewer 1 Report

The authors have provided a review about anti-SARS-CoV-2 strategies and the potential role of miRNA 2 in the assessment of COVID-19 morbidity, recurrence, and 3 therapy.

The work is very well organized, informative, and with outstanding figures

Reviewer 2 Report

Overview and general recommendation:

In this review the authors have listed hundreds of miRNAs that related to COVID-19, aim to find effective biomarker for clinical, however, there is no one reference on describing the sensitivity and specificity of miRNAs in plasma with COVID-19 patients. It is not good enough to discuss the section of potential therapeutic strategies towards COVID-19 from line 146 of Page 4 to line 257 of Page 6. We believe that the use of dexamethasone resulted in lower 28-day mortality among those who were receiving either invasive mechanical ventilation or oxygen alone in patients hospitalized with COVID-19 according to the reference (N Engl J Med 2021; 384:693-704. DOI: 10.1056/NEJMoa2021436). Thus, this review should be considered to add more references about some drugs on mortality in COVID-19 patients. Besides, this review should list 2-3 specific miRNAs that can help diagnose and prognosis with COVID-19 patients in parts of both Abstract and conclusion.

2.1. Major comments:

1.    In 5.5.1. Applied miRNAs therapies part (line 600 to line 617 of Page 15), there is no reference or discussion on miRNAs that related to COVID-19 disease but only H1N1 flu infection and HCV infection. Suggest you delete these data on this issue.

Reviewer 3 Report

Article is an exhaustive review of Sars-Cov2 pathology and possible therapeutic targets. Except for some minor corrections, it is acceptable for publication

  1. On pg2, line 47 , authors mention "until now (may 2021),no effective therapies have been proven". In my opinion the FDA has approved remdesevir and REGN-COV2 as therapies
  2. Pg4, line 165; should read "...in several ongoing clinical trials is still being tested"
  3. Pg6, line 275 and pg7 line 280: The wording completely changes the meaning. The current vaccines are effective against the new strains just not as much. The sentence should read " The antibodies produced by people vaccinated with Pfizer were about 80% effective against some mutations in B.1.617." Similarly line 280 should read "...vaccine, they found that the antibodies were about 67% effective against variant B.1.617"
  4. Pg 14, line 582: "First" is misspelled. 
